# Validity and Reliability of a Load Cell Sensor-Based Device for Assessment of the Isometric Mid-Thigh Pull Test

**DOI:** 10.3390/s23135832

**Published:** 2023-06-22

**Authors:** Raynier Montoro-Bombú, Beatriz Branquinho Gomes, Amândio Santos, Luis Rama

**Affiliations:** University of Coimbra, Research Unit for Sport and Physical Activity (CIDAF), Faculty of Sport Sciences and Physical Education, 3040-248 Coimbra, Portugal; beatrizgomes@fcdef.uc.pt (B.B.G.); acupidosantos@gmail.com (A.S.); luisrama@fcdef.uc.pt (L.R.)

**Keywords:** force, strength, rate of force development, peak force, force-time curve

## Abstract

In recent years, there has been an exponential increase in the number of devices developed to measure or estimate physical exercise. However, before these devices can be used in a practical and research environment, it is necessary to determine their validity and reliability. The purpose of this study is to test the validity and reliability of a load cell sensor-based device (LC) for measuring the peak force (PFr) and the rate of force development (RFD) during the isometric mid-thigh pull (IMTP) test, using a force plate (FP) as the gold standard. Forty-two undergraduate sport science students (male and female) participated in this study. In a single session, they performed three repetitions of the IMTP test, being tested simultaneously with an LC device and a Kistler force platform (FP). The PFr and RFD data were obtained from the force-time curve of the FP and compared with the LC data, provided automatically by the software of the device (Smart Traction device©). The mean difference between the results obtained by the LC device and the gold-standard equipment (FP) was not significantly different (*p* > 0.05), for both PFr and RFD, which suggests the validity of the ST results. Bland–Altman analysis showed a small mean difference in PFr = 1.69 N, upper bound = 47.88 N, and lower bound = −51.27 N. RFD showed that the mean difference was −5.27 N/s, upper limit = 44.36 N/s, and lower limit = −54.91 N/s. Our results suggest that the LC device can be used in the assessment of the isometric-mid-thigh-pull test as a valid and reliable tool. It is recommended that this device’s users consider these research results before putting the ST into clinical practice.

## 1. Introduction

In contemporary sports, prescribing the appropriate amount of force production is a crucial tool for achieving performance goals. The isometric force is one of the manifestations of force that has gained popularity in recent years. This type of force manifestation is widely used in the sports context and is characterized by working muscles under conditions that do not cause changes in muscle length. The isometric mid-thigh pull (IMTP) test is a specific strength test that quantifies peak isometric force, rate of force development, and specific force values [1], considering the force-time curve data based on the vertical ground reaction force (GRF). Previous research has used the IMTP test for the assessment of the peak force (PFr), both with athletes [1,2] and persons undergoing rehabilitation [3], to measure the rate of force development (RFD) [4], as well as the relationship of IMTP with jumping test performance [1,5]. This test is advantageous over other maximal repetition strength tests because it does not produce excess fatigue, needs fewer technical requirements [6], generally requires less time, and has a much lower risk of injury compared to a maximal dynamic repetition test [7]. The IMTP test was previously investigated, demonstrating an excellent level of reliability [7]. It is characterized by placing a barbell in the middle of the performer’s thigh [8], and the performer attempts to produce the maximum amount of isometric force possible [1]. The angles of the hip and knee used have varied in different studies [1,9], and although systematic reviews have attempted to find the most reliable execution angles [7] and create normative data for the IMTP test [10], these topics need further research. One of the limiting aspects of performing this test in the field setting is that it requires a force plate (FP) [3,8,11] These FPs are often expensive and require specialized personnel and are time-consuming in data analysis. Hence, the need for other technological resources that provide the same results with a high level of reliability and are less expensive is needed.

There are precedents in the validation of different devices for measuring muscle strength, with excellent and good reliability results [12,13,14]. However, one study has not confirmed the device’s validity, possibly due to procedure errors [6]. One of the available devices on the market is a load cell sensor-based (LC) device, the Smart Traction© (ST) (Figure 1). According to the manufacturer’s reports, the load cell ST can be used independently or in combination with overload machines and elastic bands. It is intended for recording PFr and power-related parameters such as RFD. The ST comes with software that allows the creation of athlete/patient profiles to track their training history and evaluations. It also displays real-time training through the force-time curve. If assessment is performed unilaterally, the device is designed for isometric and traction tests that allow comparisons between the left and right leg (calculating lateral strength deficits) and agonist/antagonist relationship. Although the device is offered as a practical and easy-to-use tool in field conditions, which would facilitate the context of strength evaluation for coaches, physiotherapists, and the public, we have not yet found in the available literature or within the virtual platform of the company (https://neuroexcellence.pt/ accessed on the 23 April 2023) a scientific study showing its validity and reliability.

In this sense, this study aimed to test a specific load cell sensor-based device’s (Smart Traction©) validity and concurrent reliability to estimate PFr and RFD during the IMTP test using a research-grade FP as the gold standard for GRF assessment. For this purpose, the load cell signal was used to estimate the ground reaction force. Our hypothesis stated that the experimental condition could reach acceptable levels of relative validity for the ST device and could be a valid and reliable device for measuring PFr and RFD during the IMTP test.

## 2. Materials and Methods

### 2.1. The Experimental Approach to the Problem

A repeated measures experimental design was developed to test the validity and reliability of the ST device for measuring PFr and RDF during the IMPT test. All participants performed three repetitions of the IMPT test with the ST device and simultaneously with a calibrated Kistler FP. The same evaluator performed all experimental evaluations.

### 2.2. Subjects

Twenty-eight men and fourteen women, all university students of sports science, were recruited for this study (24.7 ± 6.3 years old; weight 76.5 ± 10.27 kg; height 1.77 ± 0.06 m; and BMI 23.8 ± 3.1 kg·m^−2^). To be included in the study, the students had to have no contraindications to performing physical exercise and not have suffered injuries in the prior six months. Participants were instructed not to drink alcoholic beverages and to avoid intense physical activity 48 h before the evaluation session. All participants were fully informed of the associated risks and familiarized with the experimental procedures before participating. The research was conducted following the ethical principles of the Declaration of Helsinki 2010 [15] and approved by the ethics committee of the Faculty of Sport Sciences and Physical Education of the University of Coimbra (code: CE/FCDEF-UC/00802021).

### 2.3. Procedures

The experiment began with a 10 min joint mobility warm-up, followed by 5 min continuous low-intensity running, and, after, dynamic stretching for between 4 and 5 min. As part of the warm-up, a familiarization protocol involving three submaximal attempts at progressively increasing intensities in the IMTP test was included during the test session. Only the three maximal attempts, for each subject, performed after the warm-up were considered for data analysis. The test required participants to place their feet approximately hip-width apart and over a line previously set at the FP y axes (Fy). All participants held the bar with two hands so that ST was between their knees and maintained a performing position the same as in previous studies (Figure 2). The knee angle was positioned at 127 to 145° [11,16], and the hip angle was sought to be acute and between 135 and 140° [17,18] to avoid backward movements and keep the body aligned, as much as possible, with the vertical axis. A goniometer verified these angles. It was also validated in each attempt that there were no differences in the distance from the bar to the ground for each subject and that they did not throw themselves backward during the execution of the IMTP test. Participants were advised to perform the movement with minimal tension (less than 20 N) and then pull the bar vertically as hard and quickly as possible. The following three maximal attempts with a time of 5 s were then recorded for data analysis [11]. Verbal stimuli (ready, go, go, go, go, go, go, go, go stop) were offered throughout the assessment.

The FP data analysis was performed following previous recommendations [11]. The PFr was considered from when 20 N was exceeded after the onset of the action until the PFr within the force-time curve, and the RFD was measured up to 100 ms using the equation RFD = ∆force/∆time [19,20]. PFr and RFD data from the ST were reported by the system software’s device (Figure 3).

### 2.4. Instruments

A support unit was developed consisting of a wooden steel base plate with two lateral steel ropes (adjustable) that held the ST device so that it was aligned between the two knees of the subjects. This steel bar was 8 mm thick and 40 cm long. The steel lateral ropes had an adjustment system that allowed for adjusting the angles required to evaluate each subject. On this support unit, a FP (Kistler Model 9260AA6, Kistler, Winterthur, Switzerland) (Figure 1) was placed to allow the simultaneous collection of data, which was displayed at a sampling rate of 125 Hz using an interface box (DAQ system (USB 2.0), Type 5691A1, Winterthur, Switzerland) and was processed with the Bioware version 5.3.2.9 software (Winterthur, Switzerland) following the manufacturer’s instructions. Simultaneously, the ST device (Neuroexcellence, Braga, Portugal) also recorded data at 125 Hz through its Nexso App (Neuroexcellence, Braga, Portugal).

The ST is a device that uses load cells for bi-directional tension and compression and can read up to 500 kg (5000 N) of force with a sensitivity of 1.0 ± 20% mV/V, non-linearity of 1% F.S., and repeatability of 0.5% F.S. It has a rechargeable lithium polymer battery with a charge cycle of 1.5 h and a power consumption of 50 mA during charging. It measures 50 mm high, 70 mm wide, 110 mm deep, and weighs 270 g. This device is compatible with iOS 12 and above, or Android 7.0 and above using Bluetooth 4.2 and the above transmission protocol. The ST was calibrated according to the manufacturer’s specifications, and the software was installed on an iPad 10 Pro, meeting the prerequisites for use.

### 2.5. Statistical Analysis

The Shapiro–Wilk test identified a normal distribution among the data, which allowed the use of a parametric statistic. A priori statistical power was determined based on the difference between two dependent means (pairwise). A beta value of 95% was achieved, with an alpha of 0.05 and a moderate effect size (0.6), requiring a sample size of 32 participants. The validity was determined following the same strategy as previous reports [21,22]: (1) mean differences using the *t*-test, (2) Bland–Altman analysis, and (3) a calculation of the intraclass correlation coefficient (ICC). As a first indicator of validity, the one-sample *t*-test was used to test whether the difference in measurements reported by the two devices was not statistically significant. A Bland–Altman plot was used to test the level of agreement between devices. A simple linear regression model was run to check for possible biases between the two devices. The dependent variable was the mean difference, and the independent variable was the mean of the two measurements. The level of significance was established (*p* ≤ 0.05). The ICC was used to determine the level of agreement between ST and FP. The following ICC thresholds used in previous reports [21,22] were considered: poor (<0.5), moderate (0.51–0.75), good (0.76–0.9), and excellent (>0.9). Following Cohen [23], the strength of the correlations was considered as follows: r = 0.00–0.10 was considered trivial, r = 0.11–0.30 was considered minor, r = 0.31–0.50 was considered moderate, r = 0.51–0.70 was considered large, r = 0.71–0.90 was considered very large, and r = 0.91–1.0 was considered almost perfect. To report possible measurement error, the standard error of the mean (SEM) was calculated, along with the coefficient of variation (CV), which allowed for the testing of acceptable absolute reliability following previous recommendations (CV > 10% = poor, 5–10% = moderate, <5% = good) [24]. Absolute agreement was chosen for the test–retest study design [25,26]. Alpha was set at *p* ≤ 0.05. Data were analyzed with the statistical package IBM SPSS Statistics (version 27; IBM, Chicago, IL, USA), with G*Power software (v.3.1.9.7 Heinrich-Heine University of Düsseldorf, Germany), and graphs were produced with GraphPad (version 9.4.0., GraphPad Software; Boston, MA, USA).

## 3. Results

All data are presented as mean ± SD, standard error of the mean (SEM), and coefficient of variation CV (if indicated). For the PFr, the ST device showed a mean of 1473 ± 327 N, while for the FP the mean was 1475 ± 331 N. For the RFD, the ST device presented a mean of 6689 ± 2236 N·s^−1^, while, with the FP, the mean value was 6766 ± 2203 N·s^−1^. For both variables, the *t*-tests showed that the mean difference (MD) was not statistically different to zero (*p* > 0.417), thus evidencing a concordance between both devices.

The Bland–Altman analysis showed that 98.2% of the data were within the limit of agreement (LOA). Figure 4A, comparing the PFr between both devices, shows an MD = −1.69 N, with an upper limit (UL) = 47.88; lower limit (LL) = −51.27. On the other hand, Figure 4B, comparing the RFD, showed an MD = −5.27 N with UL = 44.36: LL = −54.91.

Both devices showed no proportional bias for any of the variables analyzed: PFr (*p* = 0.833, SEM = 29.16, CV = 6.3%) and RFD (*p* = 0.489, SEM = 196.6, CV = 8.1%). These results suggest that the values are normally distributed above and below the mean (Figure 5A,B). Finally, an excellent ICC was also reported in PFr (ICC = 0.96, CI = 0.92–0.99) and RFD (ICC = 0.98, CI = 0.94–0.99).

## 4. Discussion

This study aimed to test the concurrent validity and reliability of the variables PFr and RFD obtained when performing the IMTP test, measured by the ST device and compared with an investigational grade FP as the gold standard. This is the first scientific validation study with the ST device. Considering all the collected data, 98.2% was within the limits of the agreement. Moreover, the device does not show significant differences between means, and the values are correctly distributed above and below the mean. The ICCs obtained values are considered excellent [23], demonstrating that the ST device is reliable for measuring PFr and RFD during the IMTP test. Our results are consistent and in close agreement with other studies that presented similar evaluative procedures [6,12,13,14]. These studies are helpful and justified in scientific research as more and more devices are made to support the practice of physical activity and sports. A rigorous validation should be conducted before these devices start to be commercialized and used to assess performance. It is also beneficial for researchers to quantify the measurement error to ensure that inferences can be interpreted within the limits of the equipment used [22].

The concordance analysis between the devices showed no statistical difference from zero, which indicated the first step to state that both variables were valid. *T*-tests can be understood as the simplest analysis of concordance between methods, but other investigations of the same type have preferred to dispense their use [6,27]. The Bland–Altman analysis for both PFr and RFD showed that the mean differences are close to zero (PFr = −1.69, RFD = −5.27), but both have negative values. These results are consistent with other studies presenting similar values in their means [14,28]. In addition, the LOA is higher than in some other studies [13,14] but lower than in other studies that also presented excellent validity [28]. This amplitude in the LOA, although it could be expected, is typical in this kind of study design. When the IMTP is performed with devices such as ST, where the bar is in a free position that facilitates the anterior–posterior and mediolateral movement of the body in space [6], it is necessary to make sure that the movement always occurs in the vertical axis. In this way, it can be guaranteed that the FP forces are not dissipated in anteroposterior or mediolateral directions, since in opposition, the load cell will always pick up the same force in any direction of movement, which could cause an overestimation of the load cell results compared with the ones on FP. In this regard, reducing the hip angles is recommended to reduce the front-to-back motion. Our study was checked to ensure that the forces produced by the participants were produced along the vertical axis. The LOA of the ST for both PFr and RFD was approximately 50 N, which ratifies its validity for evaluating these variables.

Based on previous criteria [25], the ICC found in the study demonstrates the excellent reliability of the ST device for the measurement of PFr and RFD. A previous study [22] recommended not only looking at the ICC value but also at confidence intervals, which sometimes range from poor to excellent [6], and this may limit the ICC criteria. However, other studies have excellent confidence intervals, which is evidence of the overall reliability of the devices [27,28]. In our study, the confidence intervals had excellent PFr (ICC = 0.96, CI = 0.92–0.99) and RFD (ICC = 0.98, CI = 0.94–0.99), which ratifies the reliability of the TS device. The results of the simple linear regression (Figure 4A,B) show that the device, for none of the two variables, showed a trend to overestimate or underestimate values, and all were correctly distributed both above and below the mean, a result that also corroborates the agreement between the two methods.

Finally, these results confirm that a load cell sensor-based device is valid for estimating PFr and RFD during the IMTP test. Although our study to validate the RFD and FPr metrics used a strategy to analyze the data within the same repetition, it is recommended to consider the results of previous research [29], which suggest that when seeking greater efficiency in measuring FPr force, the assessment should be separated from RFD (whenever possible). In addition, the instructions in both situations should be independent, i.e., as “hard” as possible for FPr force or as “fast” as possible for RFD [29]. Likewise, considering that the device presents a high level of validity, the principle of operation of the load cell should be the same regardless of its horizontal or vertical use. This is an essential practical finding, especially considering the possibility of providing information to help clinicians and researchers plan treatment/rehabilitation, assess progress, manage cases, and rate impairment.

Although our results are consistent and showed the validity of the ST load cell sensor-based device for IMTP measurement, they also present some limitations regarding the data collection rate. The factory reports present a sampling rate of 125 Hz, but we verified that this has a variation of ±2 Hz, which could influence the quality of the results. A similar occurrence could happen with the ST equipment. At the same time, we hypothesize that the steel and wood platform could be improved, ensuring a rigorous setup to produce reliable results with other devices and muscular actions.

## 5. Conclusions

The load cell sensor-based device, i.e., the ST, can provide valid and reliable PFr and RFD measurements when the IMTP test is performed. The measurement error is very low compared to the minor change worth mentioning, and so ST can also be considered a sensitive instrument to detect variations in performance. This could have great practical applications for strength and fitness coaches, because as with any iOS or Android smartphone, they can access these measurements. The ST is inexpensive and cost-effective compared to piezoelectric platforms or other isokinetic force systems that may be out of reach for trainers, despite being considered the gold standard.

## Figures and Tables

**Figure 1 sensors-23-05832-f001:**
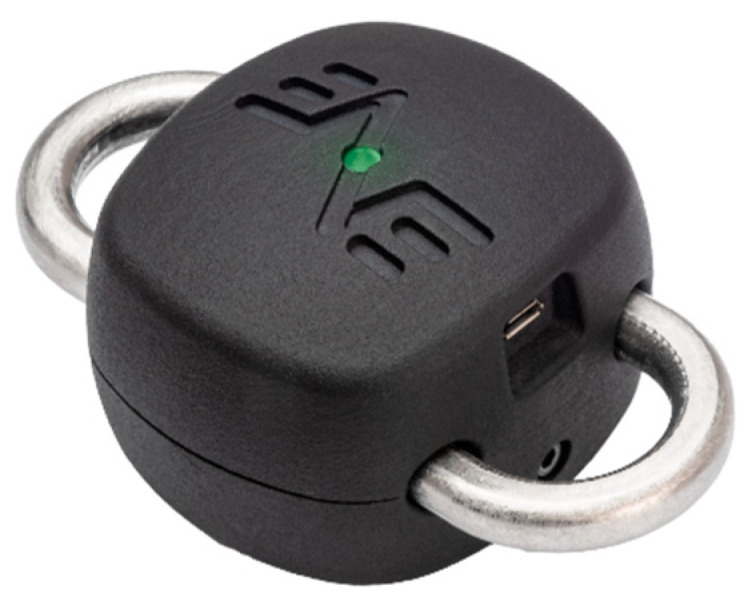
Smart Traction developed by the company Neuroexcellence.

**Figure 2 sensors-23-05832-f002:**
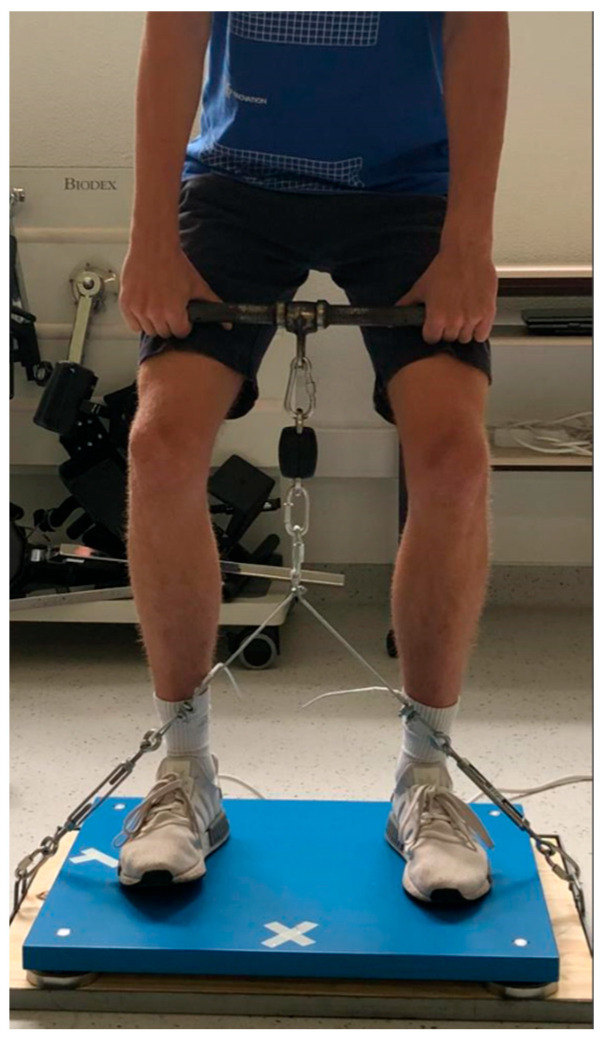
Placement of the Kistler platform on the system developed for this study.

**Figure 3 sensors-23-05832-f003:**
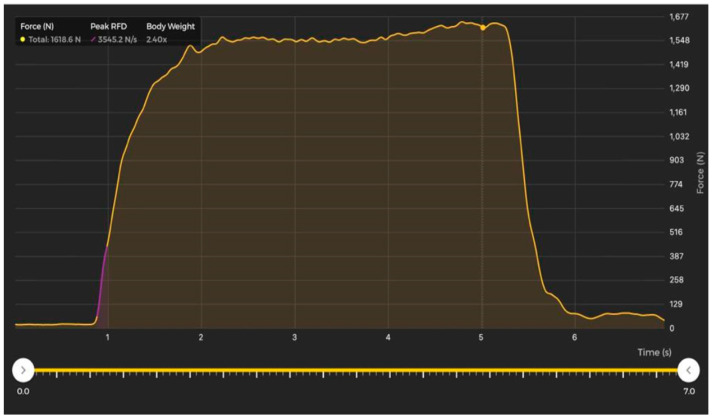
Capture of the force-time curve during IMTP provided by ST device.

**Figure 4 sensors-23-05832-f004:**
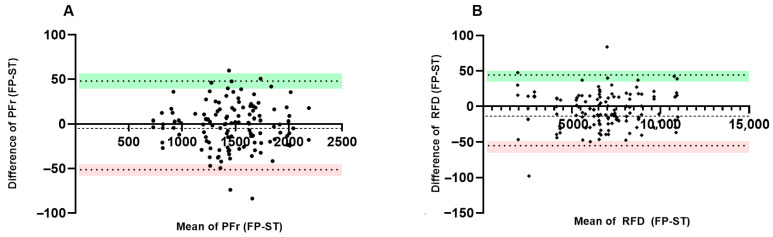
Bland–Altman test. Measurements are obtained from the FP and ST. Representation (**A**) difference of mean of PFr vs. mean PFr. Representation (**B**) difference of mean of RDF (FT) vs. mean RFD. The green color represents the upper limit (UL), while the pink color represents the lower limit (LL).

**Figure 5 sensors-23-05832-f005:**
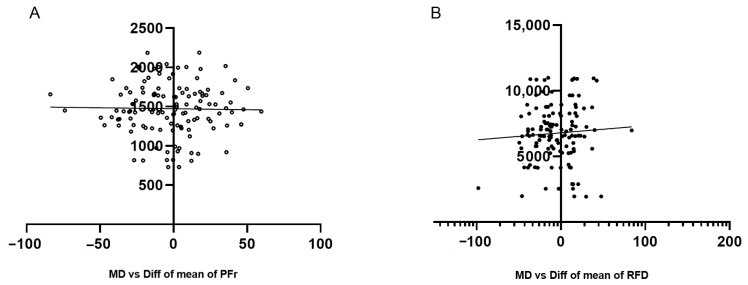
Simple linear regression. Measures are obtained from the difference of means vs. device means. (**A**) PFr representation between ST vs. FP and (**B**) RFD representation between ST vs. FP.

## Data Availability

Not applicable.

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
