# Peer review of "Validity and Reliability of a Load Cell Sensor-Based Device for Assessment of the Isometric Mid-Thigh Pull Test"

_sensors, 2023, doi:10.3390/s23135832_

Round 1

Reviewer 1 Report

GENERAL COMMENTS

Thank you for giving me the opportunity to review the article. In general, the article is written well.

Nevertheless, my opinion is that provided comments must be taken into consideration by the author before considering the article for publication.

The comments, broken down by manuscript sections, can be found below.

Title:

Maybe change the name of the device.

The suggested name is a specific name of a product, which may not be appropriate for a scientific article. It's better to keep it more general, such as "A Load Cell Sensor-Based Smart Traction Device for IMTP Assessment. Or at least: Smart Traction device©

ABSTRACT

18 – delete on

19 – why 125 Hz? Is this necessary to mention in the abstract? I think that 1000 Hz is a gold standard. Nevertheless, if you mentioned FP Hz, then you should also mention ST frequency of the data acquisition.

21 – obtained

21 – different from zero or different between the devices? Is this the same information?

23 – what does ± represent? standard deviation? Heterogeneity of the difference is quite high.

28 – change keywords – check grammar!

INTRODUCTION

40 - (1)?

46 – what about the external validity of the test in combination with dynamic tests? This should be mentioned.

Here, I can see what was the aim of your study. Please add some information regarding your equipment in the abstract. For example introduce IMTP test, its limitations (need of force plates)… and that ST can be a solution

62 – flywheel devices

65 – add: if assessment is performed unilaterally

72 ,73 – it should be mentioned that you used load cell signal to estimate ground reaction force!

PROCEDURES

All 108 attempts were considered for data analysis >> which attempts? warm-up?

118 – what is minimal tension? objectivize!

as hard as possible >> is this ok for RFD measure? check 10.1007/s00421-016-3346-6

If I get it right, RFD and PF measures were calculated from the same “type” of repetitions. i.e. as hard as possible. Explain possible limitations of this procedure in the limitations.

Please, report pretension data (average of pretension in N/kg) if possible.

143 – an answer to my comment in the abstract. Mention simultaneously. Please, emphasize possible limitations of 125 Hz data acquisition for RFD calculation

STATISTICAL ANALYSIS

159 – which software was used? I would suggest you to use reliability/validity calculator. You can find it on internet. 10.21315/eimj2018.10.3.8

Were did you get ES 0,6? What does it mean? this has to be emphasized.

163 – which type of the ICC was used?

The dependent variable was the mean differences, and the independent 168 variable was the mean of the two measurements.

Is this ok: The dependent variable was the mean differences – maybe only difference between the devices

172 – check intervals. Numbers should not be a part of two intervals at the same time

176 – SEM - measurement

186 – was unit of CV %?

support unit is cheaper, but harder to transport

185 – data should be normalized to the participants’ body mass! Change, please.

Name of axes should be changed to, for example A: PFr difference (FP – ST); PFr mean (FP, ST). Abbreviations should be explained in the text bellow the figure. Use units of measurement when reporting the results.

 192 – where/how did you get this result? 98.2%

All measures should be normalized to body mass. Use units of measurement when reporting the results.

moreover, how was reliability calculated? how was ICC calculated? how many repetitions was included in the analysis. I assume 3, but this has to be emphasized and more written systematically trough the text to .

DISCUSSION

215 – correctly? consider rewording

229 – check for typos. Spaces etc.

253 – correctly

257 – this is not true. You estimated ground reaction force (y oz z vector from Kistler plates) not force. Future studies are required to check the validity of the device in other settings, where lever arms and pull directions are different.

The limitations paragraph is missing. Why did differences between the devices occur? Is wooden steel base plate an optimal equipment that can be used for IMTP? Explain other possibilities. Some other devices for IMTP exist on the market.

Why is your device better than other load cells?

264 - the device manufacturer's team may need to continue. Did you disclose any potential conflict of interests?

Can be improved. Check for typos and grammar.

Author Response

EVALUACIÓN 1

Thank you very much for your interesting comments and contributions to the improvement of the manuscript.  The authors appreciate the value of the comments and recognize that they were important in consolidating the paper. We will also try to satisfy each of the questions, based on a review of the content and supported by the previously published bibliography.

Title:

Maybe change the name of the device.

The suggested name is a specific name of a product, which may not be appropriate for a scientific article. It's better to keep it more general, such as "A Load Cell Sensor-Based Smart Traction Device for IMTP Assessment. Or at least: Smart Traction device©.

R/ Thanks very much for this recommendation, we have changed in the manuscript according to your suggestion.

ABSTRACT

18 – delete on

R/ Thank you very much for this comment, we have deleted it in the manuscript.

19 – why 125 Hz? Is this necessary to mention in the abstract? I think that 1000 Hz is a gold standard. Nevertheless, if you mentioned FP Hz, then you should also mention ST frequency of the data acquisition.

R/ Thank you very much for this comment, we have deleted it in the manuscript.

21 – obtained

R/ Thank you very much for this comment, we have corrected it in the manuscript.

21 – different from zero or different between the devices? Is this the same information?

R/ Thank you very much for this comment.

No, it's not the same information. A one-sample t-test was used to compare whether the measures were statistically different from zero. In this type of study, the results of the T-test can be considered the first test of concordance between devices. In our case, the mean difference between the two devices was not statistically different from zero and constituted the first validation step.  

23 – what does ± represent? standard deviation? Heterogeneity of the difference is quite high.

R/ Thank you very much for this comment. You are correct, but this was corrected in the manuscript. Heterogeneity assumptions were removed using Bland Amland analysis (mean ± 1.96 (standard deviation)) and setting the Limits of Agreement (LOA), not with the Standard Deviation.

28 – change keywords – check grammar!

 R/Thank you very much for this comment, we have corrected it in the manuscript. However, we found no reason to change the keywords.

INTRODUCTION

40 - (1)?

 R/Thank you very much for this observation, we have corrected it in the manuscript.

46 – what about the external validity of the test in combination with dynamic tests? This should

R/Thank you very much for this comment Although we recognize the importance of this commentary focused on the benefits of the dynamic strength index resulting from the combination of isometric strength with jumping exercises. The authors agree with the concreteness of information and not to open gaps that are beyond the objective of ingestion.

65 – add: if assessment is performed unilaterally

R/ Thank you very much for this comment, we have added it in the manuscript.

72 ,73 – it should be mentioned that you used load cell signal to estimate ground reaction force!

R/ Thank you very much for this comment, we have added it in the manuscript.

PROCEDURES

All 108 attempts were considered for data analysis >> which attempts? warm-up?

 R/Thank you very much for this observation, we have corrected it in the manuscript.

118 – what is minimal tension? objectivize!

R/Thank you very much for this observation. This minimum required tension was less than 20 N. In this sense, the lateral wires were taut, thus ensuring that there was no bias during the evaluation. The information concerning the 20 N was also added in the manuscript.

as hard as possible >> is this ok for RFD measure? check 10.1007/s00421-016-3346-6

If I get it right, RFD and PF measures were calculated from the same “type” of repetitions. i.e. as hard as possible. Explain possible limitations of this procedure in the limitations.

Please, report pretension data (average of pretension in N/kg) if possible.

We thank the reviewer for this comment. You are correct about the results of measuring RDF based on hard and fast, versus only fast. The Bemben [1]´s studies were well consistent in this. However, we must remember that this is an equipment validation study. In this sense, the devices to be validated should measure the same in any of the performance conditions. Nevertheless, their suggestions were addressed in the discussion session.

143 – an answer to my comment in the abstract. Mention simultaneously. Please, emphasize possible limitations of 125 Hz data acquisition for RFD calculation.

R/  We thank the reviewer for this remark.

The main criteria we took care of in the validation process were to equalise the signal acquisition rate possible for both types of equipment.

And this is done. However, we agree that making the assessment with higher Hz values should be interesting.

STATISTICAL ANALYSIS

159 – which software was used? I would suggest you to use reliability/validity calculator. You can find it on internet. 10.21315/eimj2018.10.3.8

R/ Thank you very much for this comment. Data analysis and statistical power were supported by the programs. IBM SPSS Statistics (version 27; IBM, Chicago, IL), with G*Power software (v.3.1.9.7 Heinrich-Heine University of Düsseldorf, Germany).

For the validity analysis we used the Bland-Altman analysis, which we consider to be god standard for this type of validations. The reliability analysis we used the ICCs. The ICC was used to determine the level of agreement between ST and FP. The following ICC thresholds used in previous reports were considered: poor (<0.5), moderate (0.5-0.75), good (0.75-0.9), and excellent (<0.9). Following cohen  [2], the strength of the correlations was considered as follows: r=0.00-0.10 was considered trivial, r=0.11-0.30 was considered minor, r=0.31-0.50 was considered moderate, r=0.51-0.70 was considered large, r=0.71-0.90 was considered very large, and r=0.91-1.0 was considered almost perfect.   

Were did you get ES 0,6? What does it mean? this has to be emphasized.

R/ Thank you very much for this comment. The statistical power was calculated a priori reporting a moderate effect size (0.6), which may indicate that the selected sample meets the assumptions needed for an high statistical power, highlighting the veracity of our results.

163 – which type of the ICC was used?

R/ Thank you very much for this comment. The ICC was used under the Two-way mixed model with the absolute concordance type.

The dependent variable was the mean differences, and the independent 168 variable was the mean of the two measurements.

Is this ok: The dependent variable was the mean differences – maybe only difference between the devices

R/ Thank you very much for this comment. we have corrected it in the manuscript.

172 – check intervals. Numbers should not be a part of two intervals at the same time

R/ Thanks very much for this recommendation, we have changed in the manuscript.

176 – SEM – measurement

R/ Thank you very much for this comment. SEM = standard error of the mean

186 – was unit of CV %?

R/ Thank you very much for this comment. Yes, you are right.

185 – data should be normalized to the participants’ body mass! Change, please.

R/ Thank you very much for this comment. This approach may be debatable. normalization of data to weight in some comparison studies may be useful. But this precedent for validation studies may lead to bias in the results. With respect we think that absolute values are preferable in this analysis.

Name of axes should be changed to, for example A: PFr difference (FP – ST); PFr mean (FP, ST). Abbreviations should be explained in the text bellow the figure. Use units of measurement when reporting the results.

R/ Thanks very much for this recommendation, we have changed in the manuscript.

moreover, how was reliability calculated? how was ICC calculated? how many repetitions was included in the analysis. I assume 3, but this has to be emphasized and more written systematically trough the text to .

R/ Thank you very much for this comment. The ICC was used under the Two-way mixed model with the absolute concordance type. Three attempts were analyzed. This was added to the procedure.

DISCUSSION

215 – correctly? consider rewording

R/ Thanks very much for this recommendation, we have changed in the manuscript.

229 – check for typos. Spaces etc.

R/ Thanks very much for this recommendation, we have changed in the manuscript.

253 – correctly

257 – this is not true. You estimated ground reaction force (y oz z vector from Kistler plates) not force. Future studies are required to check the validity of the device in other settings, where lever arms and pull directions are different.

R/ Thanks very much for this recommendation. We have added a small modification to the text. It should also be remembered that the principle of operation of a load cell is to measure the energy that is applied to the inputs and the tension is taken from the outputs. The magnitude of the voltage depends on the load applied to the measuring sensor and this is translated into force, regardless of the type of force applied and the axis of execution.

The limitations paragraph is missing. Why did differences between the devices occur? Is wooden steel base plate an optimal equipment that can be used for IMTP? Explain other possibilities. Some other devices for IMTP exist on the market.

R/ Thanks very much for this recommendation. we have included a section with the limitations that we consider most relevant to the study.

Why is your device better than other load cells?

 R/ Thanks very much for this recommendation. Is beyond the scope of the research, the comparison with other studies.

264 - the device manufacturer's team may need to continue. Did you disclose any potential conflict of interests?

 R/ Thanks very much for this recommendation. Conflicts of Interest: The authors declare no conflict of interest.

References

  1. Bemben, M.G.; Clasey, J.L.; Massey, B.H. The Effect of the Rate of Muscle-Contraction on the Force-Time Curve Parameters of Male and Female Subjects. Research Quarterly for Exercise and Sport 1990, 61, 96-99, doi:Doi 10.1080/02701367.1990.10607484.
  2. Cohen, J. Statistical power analysis for the behavioral sciences, 2nd ed.; L. Erlbaum Associates: Hillsdale, N.J., 1988; pp. xxi, 567 p.

Reviewer 2 Report

Well written.  I found things to comment on from the Intro section

Introduction

1st paragraph: The second sentence could benefit from revision into 2 sentences

This sentence isn’t clear.  What is the isometric one? One manifestation of force that has gained popularity in recent years is the isometric one

And it’s a test, not a manifestation of force

IMTP – what does the I stand for?

RFD – needs to be spelled out prior to abbrev

Performance risk?  Wouldn’t that be risk of injury?

Needs revision for grammar: One of the limiting aspects of performing this test in the field setting is that it requires a force platform (FP) [3,8,11] that is mainly a laboratory equipment, being expensive and with low portability

However, one study have not confirmed the devices validity, possibly due to procedure errors

However, one study has not…. This sentence could benefit from revision for grammar

Some English grammar that needs to be fixed in the intro

Author Response

 Thank you very much for your interesting comments and contributions to the improvement of the manuscript.  The authors appreciate the value of the comments and recognize that they were important in consolidating the paper. We will also try to satisfy each of the questions, based on a review of the content and supported by the previously published bibliography.

Well written.  I found things to comment on from the Intro section

Introduction

1st paragraph: The second sentence could benefit from revision into 2 sentences

R/ Thanks very much for this recommendation, we have changed in the manuscript.

This sentence isn’t clear.  What is the isometric one? One manifestation of force that has gained popularity in recent years is the isometric one.

R/ Thanks very much for this recommendation, we have changed in the manuscript.

And it’s a test, not a manifestation of force

R/ Thanks very much for this recommendation, we have changed in the manuscript.

IMTP – what does the I stand for?

R/ Thanks very much for this recommendation, we have changed in the manuscript.

RFD – needs to be spelled out prior to abbrev

R/ Thanks very much for this recommendation, we have changed in the manuscript.

Performance risk?  Wouldn’t that be risk of injury?

R/ Thanks very much for this recommendation, we have changed in the manuscript.

Needs revision for grammar: One of the limiting aspects of performing this test in the field setting is that it requires a force platform (FP) [3,8,11] that is mainly a laboratory equipment, being expensive and with low portability

R/ Thanks very much for this recommendation, we have changed in the manuscript.

However, one study have not confirmed the devices validity, possibly due to procedure errors

However, one study has not…. This sentence could benefit from revision for grammar

R/ Thanks very much for this recommendation, we have changed in the manuscript.

Round 2

Reviewer 1 Report

Thank you for providing a response to my comments. Nevertheless, it can be seen that not all of my comments were addressed. You stated that your answers are supported by the literature, but this is not true. The majority of your answers are assumptions, as you stated. For example, I would like to know why normalizing the data to body mass can lead to higher bias than using non-normalized data, etc.

These suggestions should at least be mentioned in the limitations paragraph.

Author Response

Reviewer 1

Thank you for providing a response to my comments. Nevertheless, it can be seen that not all of my comments were addressed. You stated that your answers are supported by the literature, but this is not true. The majority of your answers are assumptions, as you stated. For example, I would like to know why normalizing the data to body mass can lead to higher bias than using non-normalized data, etc.

These suggestions should at least be mentioned in the limitations paragraph.

R/ Again, thank you very much for your comments. Your contributions are very important to improve our manuscript.  In our first round of responses to reviewers' comments, we tried to do our best to correspond to every issue pointed. We refer to the literature to support our statement whenever we thought that it was adequate for the purpose.

Concerning the normalization of force data, we join an explanation that supports our decision We understand your concerns however, for the aim of this study, the authors think that using non-normalized force data (raw data) is suitable.

Different researchers have normalized data to body weight when analyzing strength variables (1, 2). However, others had discussed this issue previously, highlighting the inconsistencies and the potential risks of the use of normalization to body weight. An example is the study of Wannop JW, and col. (3) which showed that this method only accounts for some of the variability if the intercept is equal to zero, which is usually not the case. In addition, O'Malley MJ states that If the intercept is different from -zero, the result would be a normalized GRF value GRF/BW, with a residual relationship to BW due to the new b/BW term (1).

Finally, considering that we do not know with certainty whether this normalization eliminates all variability, and we also do not know whether the forces recorded when divided by body mass may be overestimated or underestimated or whether the values being compared are due solely to condition and not to differences between subjects, the authors prefer to use the net GRFs net values shown by the equipment. These forces (N) do not interfere with the investigation results.  In our case, we assume the strategy to analyze the results in N, as reported in other validation studies that do not use normalized values by weight (5, 6, 7).

We hope that our explanation could be accepted as it will not affect the propose of a validation study

References

  1. Jensen RL, Ebben WP. Quantifying plyometric intensity via rate of force development, knee joint, and ground reaction forces. Journal of Strength and Conditioning Research. 2007;21(3):763-7.
  2. Besier TF, Lloyd DG, Cochrane JL, Ackland TR. External loading of the knee joint during running and cutting maneuvers. Med Sci Sports Exerc. 2001;33(7):1168-75.
  3. Wannop JW, Worobets JT, Stefanyshyn DJ. Normalization of ground reaction forces, joint moments, and free moments in human locomotion. J Appl Biomech. 2012;28(6):665-76.
  4. O'Malley MJ. Normalization of temporal-distance parameters in pediatric gait. J Biomech. 1996;29(5):619-25.
  5. Cramer LA, Wimmer MA, Malloy P, O'Keefe JA, Knowlton CB, Ferrigno C. Validity and Reliability of the Insole3 Instrumented Shoe Insole for Ground Reaction Force Measurement during Walking and Running. Sensors (Basel). 2022;22(6).
  6. Lodge C, Tobin D, O'Rourke B, Thorborg K. Reliability and Validity of a New Eccentric Hamstring Strength Measurement Device. Arch Rehabil Res Clin Transl. 2020;2(1):100034.
  7. Baena-Raya A, Diez-Fernandez DM, Garcia-Ramos A, Soriano-Maldonado A, Rodriguez-Perez MA. Concurrent validity and reliability of a functional electromechanical dynamometer to assess isometric mid-thigh pull performance. Proceedings of the Institution of Mechanical Engineers Part P-Journal of Sports Engineering and Technology. 2021.